# Expression of Transferrin Protein and Messenger RNA in Neural Cells from Mouse and Human Brain Tissue

**DOI:** 10.3390/metabo12070594

**Published:** 2022-06-26

**Authors:** Eriko Abe, Takashi J. Fuwa, Kyoka Hoshi, Takashi Saito, Takenobu Murakami, Masakazu Miyajima, Norihiro Ogawa, Hiroyasu Akatsu, Yoshio Hashizume, Yasuhiro Hashimoto, Takashi Honda

**Affiliations:** 1Department of Biochemistry, Fukushima Medical University, Fukushima 960-1295, Japan; eriko-ab@fmu.ac.jp (E.A.); tjfuwa@fmu.ac.jp (T.J.F.); khoshi@fmu.ac.jp (K.H.); 2Department of Neurocognitive Science, Nagoya City University, Aichi 467-8601, Japan; saito-t@med.nagoya-cu.ac.jp; 3Division of Neurology, Department of Brain and Neurosciences, Faculty of Medicine, Tottori University, Tottori 683-8504, Japan; maaboubou@gmail.com; 4Department of Neurosurgery, Juntendo University, Tokyo 113-8421, Japan; mmasaka@juntendo.ac.jp; 5Department of Neuropathology, Fukushimura Hospital, Aichi 467-8601, Japan; norihiro@chojuken.net (N.O.); akatu@med.nagoya-cu.ac.jp (H.A.); yhashi@chojuken.net (Y.H.); 6Department of Forensic Medicine, Fukushima Medical University, Fukushima 960-1295, Japan; ponchan@fmu.ac.jp

**Keywords:** brain, choroid plexus, cortex, hippocampus, mRNA, neuron, transferrin protein

## Abstract

Iron is an essential nutrient in the body. However, iron generates oxidative stress and hence needs to be bound to carrier proteins such as the glycoprotein transferrin (Tf) in body fluids. We previously reported that cerebrospinal fluid contains Tf glycan-isoforms that are derived from the brain, but their origins at the cellular level in the brain have not yet been elucidated. In the present report, we described the localization of Tf protein and mRNA in mouse and human brain tissue. In situ hybridization of mouse brain tissue revealed that Tf mRNA is expressed by different cell types such as epithelial cells in the choroid plexus, oligodendrocyte-like cells in the medulla, and neurons in the cortex, hippocampus, and basal ganglia. In contrast, Tf protein was barely detected by immunohistochemistry in hippocampal and some cortical neurons, but it was detected in other types of cells such as oligodendrocyte-like cells and choroid plexus epithelial cells. The results showed that Tf mRNA is expressed by neural cells, while Tf protein is expressed in different brain regions, though at very low levels in hippocampal neurons. Low Tf level in the hippocampus may increases susceptibility to iron-induced oxidative stress, and account for neuron death in neurodegenerative diseases.

## 1. Introduction

Iron is an essential cofactor required for a multiplicity of biological processes throughout the body. For iron transportation, specific carrier proteins such as transferrin (Tf) in the extracellular fluid and blood is required [1]. Tf is mainly biosynthesized in the liver and secreted into blood for distributing iron to other organs including the central nervous system (CNS) [2]. The CNS, however, is separated from the systemic circulation by the blood–brain barrier, which is comprised of endothelial cells, pericytes, and astrocytic processes. Blood Tf first binds to the transferrin receptor (TfR) on endothelial cells [3]. The binding complex is internalized into endosomes. After internalization, the iron dissociates from the Tf/TfR complex in an acidic endosomal compartment. The iron-free transferrin (apotransferrin) together with TfR is then transported back to the plasma membrane where it is released into the blood to participate in further rounds of iron mobilization and delivery. Namely, Tf is recycled and never translocated into the brain parenchyma in this process. In contrast to Tf, iron in endosome is further exported to the abluminal side of the brain parenchyma by ferroportin. Exported iron immediately binds to Tf to form Tf/iron complex in the extracellular fluid of the brain. Neurons express high levels of TfR, which binds the Tf/iron complex to be internalized. After internalization, the iron is released and either stored as ferritin-bound iron or utilized for biological processes such as respiratory chain electron transfer and neurotransmitter biosynthesis; iron is essential for the initial step of catecholamine biosynthesis. Thus, the parenchymal extracellular transferrin plays a pivotal role in neural iron transport, but its origin has not been fully characterized. 

Tf is post-translationally modified with *N*-glycans, which are cell-type specific and hence indicate the cellular origin of the molecule. We recently found that cerebrospinal fluid (CSF) contains three glycan-isoforms, i.e., Tf carrying sialylα2,6 galactose-terminated glycans (Sia-Tf); Tf carrying *N*-acetylglucosamine (GlcNAc)-terminated glycans (GlcNAc-Tf) [4,5]; and Tf carrying mannose-terminated glycans (Man-Tf) [6]. Sia-Tf carries the same glycan as blood Tf, suggesting that this isoform is derived from that source. In contrast, Man-Tf and GlcNAc-Tf are derived from the cerebrum given that these isoforms were not detected in CSF of congenital hydranencephaly patients, in which the cerebrum was mostly lacking [6]. Indeed, GlcNAc-Tf is expressed in the CSF-producing choroid plexus, meaning that this isoform could serve as a biomarker for altered CSF production [5,7,8]. Man-Tf is a major isoform in the cerebral cortex (~90%) and possibly from neurons [6]. These observations suggest that Tf is biosynthesized in different regions of the brain. To this end, Tf transcript levels in the brain were previously analyzed by cDNA microarray. The analysis revealed that expression levels are different among regions of the brain [9], but detailed expression profiles at the cellular level are yet to be fully examined. In addition, regional differences in Tf protein expression have also not been clarified.

In the present study, we analyzed the expression profiles of Tf mRNA and Tf protein at a cellular level in mouse and human brain tissue, and found that Tf protein levels were very low in hippocampal neurons and some cortical neurons despite Tf mRNA being highly expressed in those cells.

## 2. Results

### 2.1. Expression of Tf mRNA in Mouse Brain 

We analyzed the expression of Tf mRNA in mouse brain slices; in situ hybridization revealed antisense probe signals in different brain areas, whereas no notable signal was detected with the sense probe (Figure 1A,B). Strong signals were detected in various brain regions such as the hippocampus, cerebral cortex, thalamic and hypothalamic nuclei, and basal ganglia. The most intense signals were observed in the choroid plexus of the lateral and third ventricles. In high magnification images of the hippocampus, intense signals were observed in the dentate gyrus as well as the CA1, CA2, and CA3 regions (Figure 1a), indicating that neurons in these regions express Tf mRNA. In the cortex, signals could be detected in the cytosolic regions of large neuron-like cells and small glia-like cells (Figure 1b). In the corpus callosum, positive signals were detected in a number of small cells (Figure 1c). Some of these formed linear rows of positive cells (asterisks), suggesting that Tf mRNA is expressed by oligodendrocytes, which form rows of cells alongside axons. Signals in the choroid plexus were detected in cytosolic regions of epithelial cells (Figure 1d), suggesting that Tf mRNA is actively produced by choroid plexus epithelial cells.

To examine the colocalization of Tf mRNA with NeuN antigen (a neuron marker) in the cerebral cortex and hippocampus, pairs of mirror-image brain sections were prepared, with one section stained with the antisense probe and methyl green, while the other was stained with anti-NeuN antibody. In the hippocampus, NeuN-positive neurons exhibited Tf mRNA signals in the dentate gyrus and CA3 region (Figure 2(A1,A2)). In the cerebral cortex, almost all NeuN-positive cells exhibited antisense signals (Figure 2(B1,B2)), suggesting that cortical neurons actively produce Tf mRNA. As far as we could determine, most NeuN-positive cells showed co-localized antisense signals, suggesting that different types of neurons express Tf mRNA, although expression levels varied across different regions of the mouse brain.

### 2.2. Expression of Tf Protein in Mouse Brain Tissue

Antibody against mouse Tf were examined by immunoblotting for their binding specificity. Serum Tf was used for the authentic standard. Mouse brain was extracted into 0.1% Triton-X 100 in phosphate-buffered saline (PBS) and the extracts subjected to SDS-PAGE followed by immunoblotting. SDA-PAGE revealed multiple bands over a wide range of molecular masses in the mouse extracts (Figure 3A). Immunoblotting showed that the anti-mouse Tf antibody reacted with a doublet band in the 72~74 kDa region (Figure 3B). Another blot probed with anti-NeuN antibody revealed that the extracts contained NeuN antigen (Figure 3C). The extracts were mixed with Tf antibody-immobilized beads to separate bound and unbound fractions. The doublet bands were detected in the bound but not the unbound fraction (Figure 3D). Doublets in the extracts migrated faster than that of serum Tf, while doublets in bound fraction migrated more slowly, possibly due to the removal of contaminant proteins.

By using anti-mouse Tf antibody, we analyzed Tf protein expression in the mouse brain by immunohistochemistry. Faint signals were detected in various brain regions (Figure 4A). To examine the colocalization of Tf and NeuN antigen in the hippocampus (a), pairs of mirror-image sections were stained with anti-Tf or anti-NeuN antibodies. High magnification images of the hippocampus revealed that neurons in the CA1, CA2, and CA3 regions as well as the dentate gyrus were stained with anti-NeuN antibody, but negligible, if any, staining with anti-Tf antibody could be seen (Figure 4(a1,a2)). When part of the dentate gyrus image was further magnified to examine signals at a cellular level, NeuN-positive neurons showed almost no staining at all with anti-Tf antibody (Figure 4(a3,a4)). This result indicated that Tf protein was virtually undetectable in the hippocampal neurons despite the apparent expression of Tf mRNA. To examine regional expression differences in the brain, the cerebral cortex was also co-stained with anti-NeuN and anti-Tf antibodies. Some NeuN-positive neurons were weakly stained with anti-Tf antibody while others showed negligible or no staining (Figure 4(b1,b2)). This observation suggested that Tf protein expression by neurons differs across different brain regions, and that Tf protein expression occurs discordantly with Tf mRNA expression. In the corpus callosum, Tf signals were detected in oligodendrocyte-like cells, some of which formed linear rows of immunopositive cells (Figure 4c, asterisks). In the choroid plexus of the third ventricle, Tf signals were detected on epithelial cells (Figure 4d).

### 2.3. Expression of Tf mRNA in the Human Cortex and Medulla

Tf mRNA expression was examined in human cortex and medulla tissue. In situ hybridization revealed antisense probe signals in the temporal lobe, whereas no signal was detected with the sense probe (Figure 5A,B). In a high magnification image of the cortex, large neuron-like cells showed NeuN immunoreactivity, with most of them also showing significant Tf mRNA signals (Figure 5(a1,a2)). This result suggests that, like mouse neurons, human cortical neurons express Tf mRNA. In the medulla (b), positive antisense signals were detected in some small cells, several of which formed in rows (asterisked) (Figure 5b), indicative of Tf mRNA-expressing oligodendrocytes. 

### 2.4. Expression of Tf Protein in Human Cortex and Medulla

Antibodies against human Tf were examined by immunoblotting for their binding specificity. Serum Tf was used for the authentic standard. Human brain was extracted into 0.1% Triton-X 100 in PBS and the extracts subjected to SDS-PAGE followed by immunoblotting. SDA-PAGE revealed multiple bands over a wide range of molecular masses in the human extracts (Figure 6A). Immunoblotting showed that the anti-human Tf antibody reacted with major and minor bands around 70~72 kDa area (Figure 6B). Immunoblotting using anti-NeuN antibody revealed that the extracts contained NeuN antigen (Figure 6C). The extracts were mixed with Tf antibody-immobilized beads to separate bound and unbound fractions. Tf antibody was detected the bands in the bound but not the unbound fraction (Figure 6D).

Next, we analyzed the expression of Tf protein in autopsied human brain tissue by immunohistochemistry. The anti-human Tf antibody revealed faint signals in the temporal lobe (Figure 7A). To examine the colocalization of NeuN antigen and Tf protein, pairs of mirror-image sections of the cortex were prepared. In a high magnification image of cortical tissue, some NeuN-positive neurons were stained with anti-Tf antibody, whereas others were negligibly stained (Figure 7(a1,a2)). This result suggested that, as for mouse neurons, human cortical neurons express different levels of Tf protein. Some oligodendrocyte-like cells in the white matter were also stained by the antibody (Figure 7b).

### 2.5. Expression of Tf mRNA and Tf Protein in Human Hippocampal Tissue

Tf mRNA and protein expression were examined in pairs of mirror-image sections of human hippocampus subjected to in situ hybridization (Figure 8(A1)). Anti-human Tf antibody staining revealed faint signals in the hippocampal region (Figure 8(A2)), while diffuse signals surrounding blood vessels were also observed. In a high magnification image, positive antisense signals were detected in CA2 neurons, which were only sparingly stained with anti-human Tf antibody (Figure 8(B1,B2)). These results suggest that human hippocampal neurons showed discordant expression of Tf mRNA and Tf protein.

## 3. Discussion

In the present study, we demonstrated that Tf mRNA is expressed in neurons of the hippocampus, and cerebral cortex of mouse and human brain tissue. Transcripts were also expressed in choroid plexus epithelium and oligodendrocyte-like cells, in which Tf protein was well expressed. In contrast to these cells, Tf protein was barely detected in hippocampal neurons and in some cortical neurons. In the case of human brain, it is possible that postmortem autolysis diminished Tf protein in these cells. However, mouse brain, minimized for postmortem changes, showed the similar Tf expression profile to human brain, suggesting that autolysis is not a major reason for the low level of the protein. Possible explanation for discordant expression of Tf mRNA and protein are as follows: (i) translational efficiency is low in these neurons, (ii) turnover of Tf protein is rapid, (iii) the mRNA is transported out from soma, via axonal flow or exosomes, for local translation elsewhere. Controlling mechanism for brain Tf levels need to be clarified in each cell-types in the future.

Tf has two metal binding pockets with extremely high affinity for ferric iron; the binding constant is 1 × 10^20^ M^−1^ at pH7.4 [10]. The high affinity is required for preventing iron-induced oxidative stress, i.e., generation of reactive oxygen species (ROS) [11]. Therefore, low expression of Tf may enhance susceptibility to iron toxicity. Indeed, iron-induced oxidative stress is hypothesized to cause Alzheimer’s disease (AD). In AD brain, neuron death and the following brain atrophy start in the hippocampal region and then spread to other cortex regions [12]. Low expression of Tf in these regions may be partly responsible for the neuron death. It is noted that the hippocampal atrophy causes short term memory-loss, which is often manifested in the initial stage of AD patients. In advanced stage, accelerated atrophy of other cortex regions leads to symptoms such as disorientation, self-neglect, and mood swings. For a new therapeutic means for these symptoms and underlying pathological changes, regional difference in iron metabolism needs to be analyzed in AD and control brain.

Glycan-isoforms show unique spatio-temporal expression patterns, which are often associated with their function. For example, Tf isoform carrying mannose-terminated glycans (Man-Tf) comprises only 8% of total Tf in cerebrospinal fluid (CSF) [7]. Man-Tf increased in CSF of AD patients, and the increment was correlated well with that of phosphorylated-tau (p-tau), which has been established as an AD-specific biomarker for neuron death. In addition, Man-Tf × p-tau index showed good diagnostic accuracy for AD. These results suggest that Man-Tf is involved in AD pathophysiology. In contrast to CSF, Man-Tf was a major Tf isoform in the cerebral cortex (~90%) and possibly derived from neurons [6]. Given that iron-induced ROS generation, a rapid chemical reaction, is involved in the neuron death, it should be prevented by local Tf, possibly Man-Tf. If not, dead neurons may further release intracellular ferritin-bound iron into the extracellular fluid, which would causes further death or injury to neighboring neurons [13]. In this context, quantification of Man-Tf in AD and control brain would be important for understanding its involvement in the pathology.

## 4. Materials and Methods 

### 4.1. Brain Tissue

Two males and 3 females of C57BL/J (16 weeks of age; CLEA Japan, Inc., Tokyo, Japan) were used. Mice were sacrificed by cervical dislocation, and brains excised rapidly. A hemisphere was used for biochemical analysis. The rest was fixed with 4% glutaraldehyde in PBS and embedded with paraffin for histochemical analysis. Human brain tissues embedded with paraffin were obtained from the autopsy patients in Fukushimura and Fukushima University Hospital. Their sex, age, and cause of death were as follows: male, 85 years old, heart failure; female, 82 years old, multiple organ failure; female, 96 years old, obstructive jaundice; male, 84 years old, respiratory failure. Human fresh brain tissue was obtained by forensic autopsy of a 51-year-old male who died from a car accident without head injury. For biochemical analysis, the fresh brain was washed three times with PBS and then proteins were extracted with PBS containing 1% TritonX-100 and 1% protease inhibitor cocktail (Nacalai Tesque, Inc., Kyoto, Japan). The protein concentration was estimated using a Micro-BCA protein assay kit (23235, Thermo Fisher Scientific-JP, Tokyo, Japan). For histochemical analysis, a pair of mirror image sections was prepared as follows: two serial sections (5 μm thick) were prepared from a paraffin block. One section was inverted and floated on PBS and then mounted on a slide glass. The other section was mounted adjacent to the inversed section on the slide glass. Two sections were separated by a hydrophobic tape and each was subjected to different staining. The representative images of a female mouse and an 82-year-old female subject were used for preparing figures, because sex-specific differences were not observed. This study was approved by the Animal Experiments Committee (approval number: animal 29037) and the Ethics Committee (approval number: 2478) of Fukushima Medical University.

### 4.2. Probe Synthesis

Digoxigenin (DIG)-labeled cRNA probes were designed on a murine or human transferrin ORF sequence (NM_133977, NM_001063.3); mouse forward; 5′-gggtaatacgactcactatagggtgcctgtgtgaagaaaacc-3′; mouse reverse, 5′-gggattaaccctcactaaagggaaactgcccgagaagaaact-3′; human forward, 5′-gggtaatacgactcactatagggctccacccttaaccaatacttc-3′; and human reverse, 5′-gtgattaaccctcactaaagggaatcccttctcaaccagacacc-3′. Forward and reverse primers were incorporated with T7 and T3 promoter sequences (underlined), respectively. The target mouse or human cDNAs (Origene, Rockville, MD, USA) were amplified by PCR. In vitro transcription was then performed using the DIG labeling system for RNA (Roche Diagnostics, Mannheim, Germany) and T3 RNA polymerase (Roche Diagnostics) for synthesizing antisense probes, or T7 RNA polymerase (Takara Bio Inc., Shiga, Japan) for synthesizing sense probes.

### 4.3. In Situ Hybridization (ISH)

DIG-labeled cRNA probes were diluted to 500 ng/mL in hybridization solution; 50% formamide, 2× saline-sodium citrate buffer, 1× Denhardt’s solution, 100 µg/mL salmon sperm DNA, and 300 µg/mL yeast tRNA (Roche Diagnostics). Mouse sections were hybridized with cRNA in a hybridization buffer overnight at 65 °C (human sections; 60 °C). The sense probe was used as a negative control. After hybridization, sections were incubated with alkaline phosphatase (AP)-conjugated anti-DIG antibody (Roche Diagnostics) or horseradish peroxidase (HRP)-conjugated anti-DIG antibody (Roche Diagnostics). Labeled RNA was detected with 4-nitroblue tetrazolium chloride (NBT)/5-bromo-4-chloro-3-indoyl-phosphate (BCIP) mixture (Roche Diagnostics) for AP and the ImmPACT DAB substrate (Vector Lab., Burlingame, CA, USA) for HRP, respectively. 

### 4.4. SDS-PAGE and Blotting Analyses

Brain tissue was washed three times with PBS containing 1% protease inhibitor cocktail. Protein in the tissue was extracted with PBS containing 1% TritonX-100 and 1% protease inhibitor cocktail. The extracts were dissolved in Laemmli sample buffer, boiled for 3 min, and loaded on a gel for sodium dodecyl sulfate-polyacrylamide gel electrophoresis (SDS-PAGE) (SuperSep Ace 7.5%, FUJIFILM Wako, Osaka, Japan). After SDS-PAGE, protein bands were visualized with a Silver Stain II kit (FUJIFILM Wako). For immunoblotting, proteins separated on gels were transferred to nitrocellulose membranes. The membranes were blocked in 3% skim milk, and incubated anti-transferrin antibodies labeled with HRP (A90-129P, anti-Mouse Transferrin; A80-128P, anti-Human Transferrin, Bethyl Laboratories, Montgomery, TX, USA). Anti-NeuN antibody was used for detecting the antigen (EPR12763, Abcam, Cambridge, UK). The membranes were developed using a SuperSignal West Dura Chemiluminescence Substrate Kit (Pierce Biotechnology, Rockford, IL, USA).

### 4.5. Immunoprecipitation of Tf in the Brain

Anti-Tf antibody was immobilized on Protein A/G agarose beads (Abcam) according to manufacturer’s protocol. Beads were incubated with brain extracts for 4 h at 4 °C, following which the supernatant was removed and the beads washed three times with PBS. Antigen-antibody complex was treated by heating the beads at 95 °C in SDS-PAGE sample buffer, which was used as a bound fraction. The supernatant was used as an unbound fraction.

### 4.6. Immunohistochemistry (IHC)

Mouse brain sections were incubated with anti-mouse transferrin (primary antibody; Bethyl) followed by anti-goat IgG-HRP (secondary antibody; Jackson Immuno Research Inc., West Grove, PA, USA). Human brain sections were incubated with anti-human transferrin antibody (DakoCytomation, Glosyrup, Denmark) followed by HRP-conjugated secondary antibody (ab205718, abcam). Signals were developed with ImmPACT DAB substrate (Vector Lab.). For double staining, DAB-stained slides were treated with quenching peroxidase and then with anti-NeuN antibody (abcam). The sections were further incubated with anti-rabbit IgG-HRP (abcam). Signals were developed with VECTOR VIP substrate (Vector Lab.).

### 4.7. Image Analysis

All images were scanned using a NanoZoomer slide scanner (HAMAMATSU, Hamamatsu, Japan) and analyzed by NDP.view2 U12388-01 software (HAMAMATSU).

## Figures and Tables

**Figure 1 metabolites-12-00594-f001:**
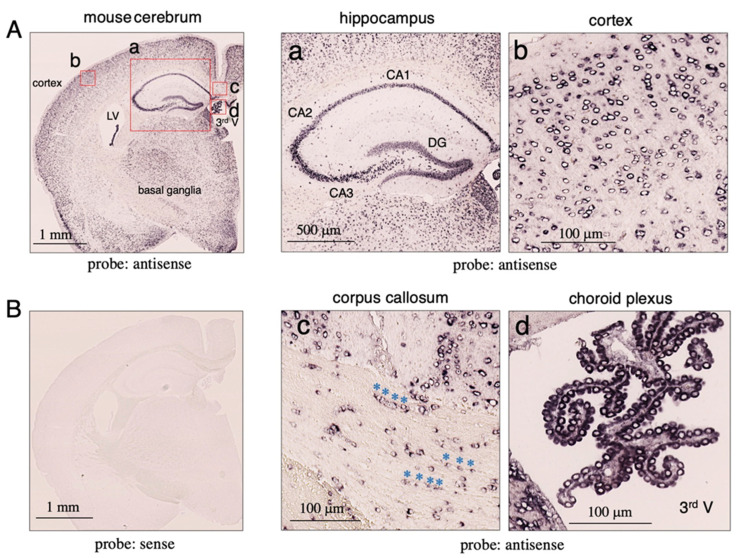
Expression of Tf mRNA in mouse brain tissue. Mouse brain sections (5 μm thick) were subjected to in situ hybridization using DIG-labeled antisense (**A**) or sense probes (**B**). The probe was detected with alkaline phosphatase-conjugated anti-DIG antibody. High magnification images of the hippocampus (**a**), cortex (**b**), corpus callosum (**c**), and choroid plexus (**d**) are shown. In the corpus callosum, Tf mRNA-positive cells are indicated with blue asterisks (**c**). The lateral (LV) and third ventricles (3rd V) are indicated (**A**).

**Figure 2 metabolites-12-00594-f002:**
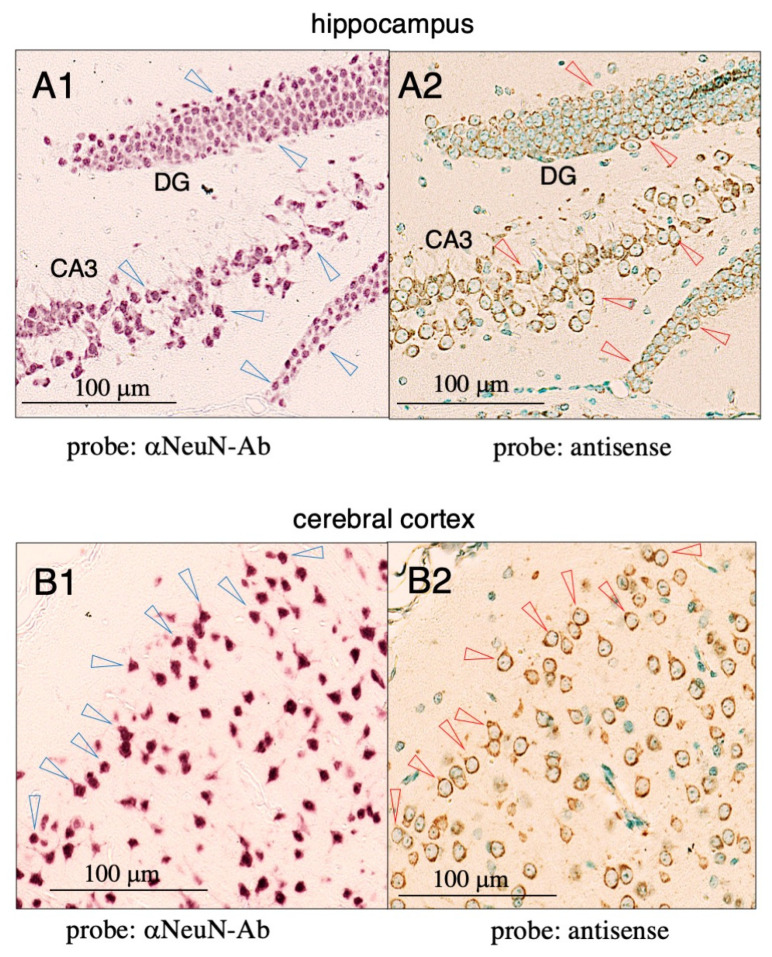
In situ hybridization of mouse cortex and hippocampus using a Tf-antisense probe and immunohistochemistry with an anti-NeuN antibody. Pairs of mirror-image sections of mouse cortex and hippocampus were prepared: one section was stained with anti-NeuN antibody for detecting neurons (**A1**,**B1**) (violet) and the other with antisense probe and methyl green for nuclear staining (**A2**,**B2**). The antisense probe was detected with horseradish peroxidase-conjugated anti-DIG antibody (brown). Some NeuN-positive cells are indicated with blue arrowheads (**A1**,**B1**) and the same positions on the mirror-image section are indicated with red arrowheads on antisense-stained sections (**A2**,**B2**).

**Figure 3 metabolites-12-00594-f003:**
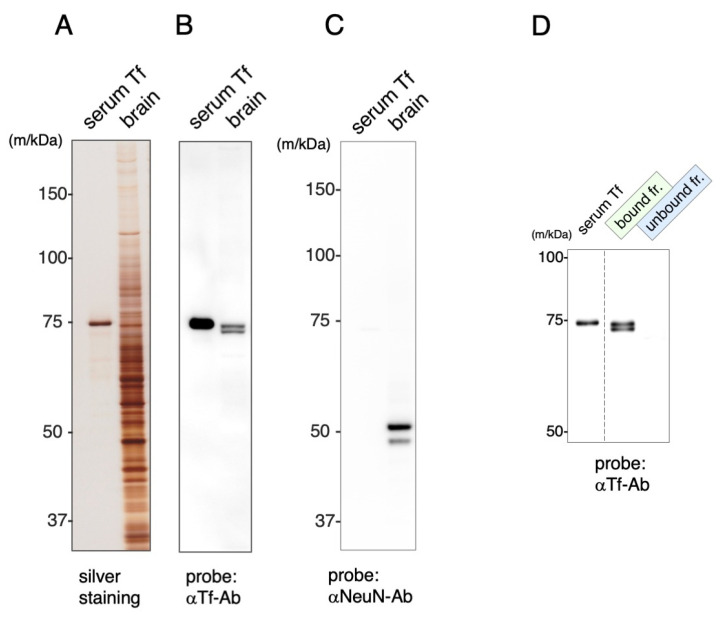
SDS-PAGE and immunoblotting of mouse brain extracts. Brain extracts were subjected to SDS-PAGE (**A**) followed by immunoblotting using anti-Tf antibody (αTf-Ab) (**B**). Another blot was probed with anti-NeuN antibody (αNeuN-Ab) (**C**). The extracts were separated into bound and unbound fractions by Tf antibody-immobilized beads and subjected to immunoblotting (**D**). Each lane separated on the gel is indicated by a dotted line.

**Figure 4 metabolites-12-00594-f004:**
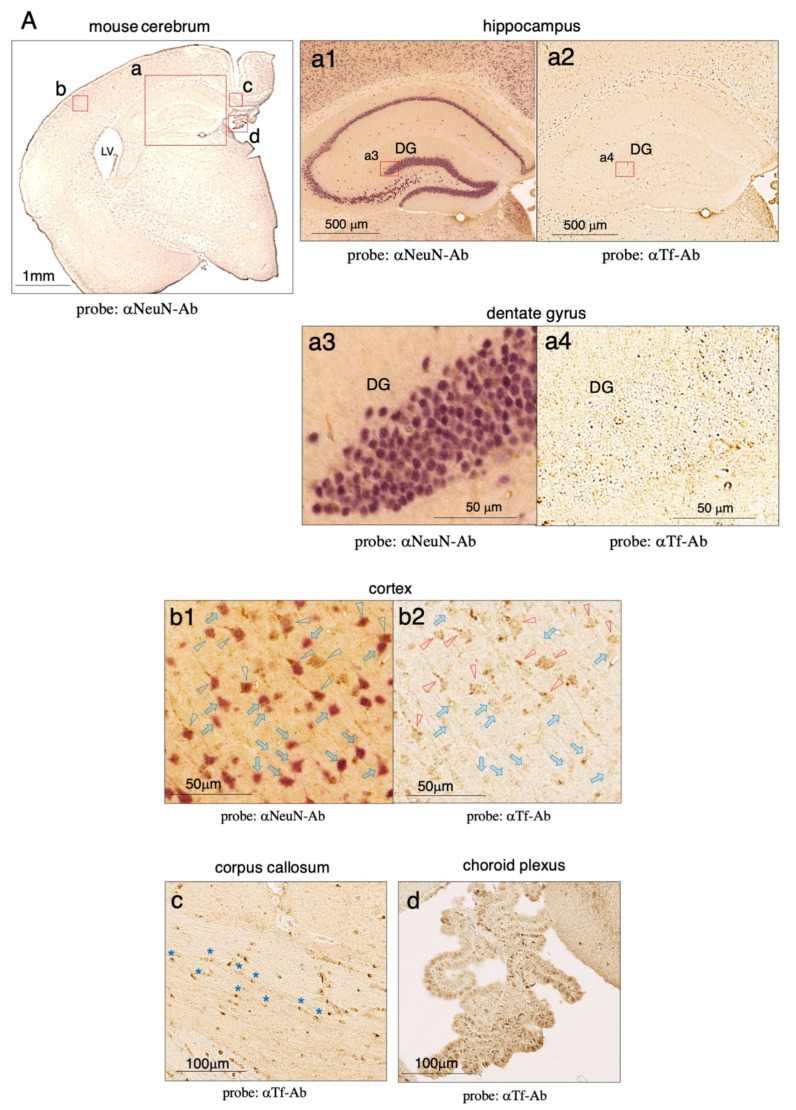
Immunohistochemistry of mouse brain sections stained with anti-Tf antibody. Mouse brain sections were subjected to immunohistochemistry using an anti-mouse Tf antibody (**A**). Lateral ventricle (LV) is shown. High magnification images of the hippocampus (**a**), cerebral cortex (**b**), corpus callosum (**c**), and choroid plexus (**d**) are also shown. Mirror-image sections of the hippocampus were stained with anti-NeuN antibody (**a1**) and anti-mouse Tf antibody (**a2**). High magnification images of the dentate gyrus (DG) are shown (**a3**,**a4**). NeuN-positive cells in the cortex are indicated with blue arrowheads (**b1**), and the same positions on the mirror-image section are indicated with red arrowheads showing Tf antibody-staining (**b2**). NeuN-positive but Tf-negative neurons are indicated with blue arrows (**b1**,**b2**). In the corpus callosum, Tf-positive cells forming rows are indicated with blue asterisks (**c**). In the choroid plexus, Tf signals can be detected on epithelial cells (**d**).

**Figure 5 metabolites-12-00594-f005:**
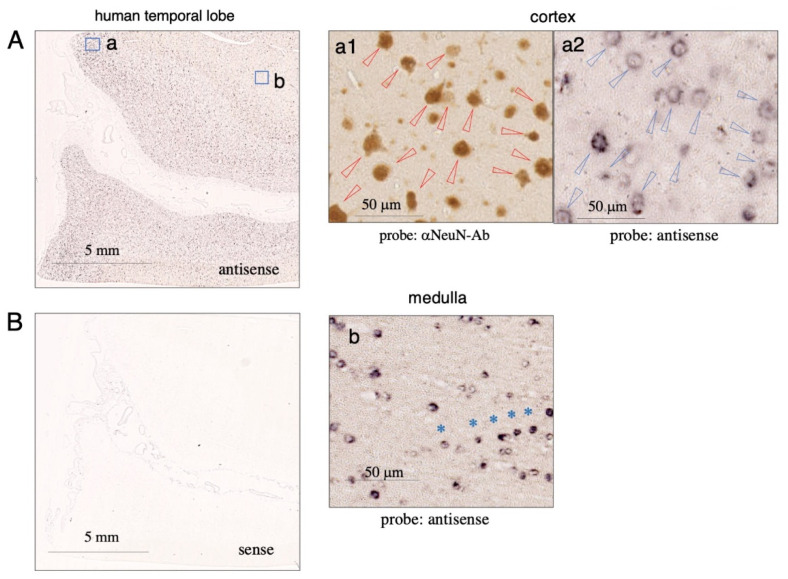
Expression of Tf mRNA in human brain tissue. Human brain sections were subjected to in situ hybridization using DIG-labeled antisense (**A**) or sense probes (**B**). The probe was detected with alkaline phosphatase-conjugated anti-DIG antibody. Pairs of mirror-image sections of cortex were prepared: one section was stained with anti-NeuN antibody for detecting neurons (**a1**) and the other with antisense probe (**a2**). NeuN-positive cells are indicated with red arrowheads (**a1**), and the same positions on the mirror-image section are indicated with blue arrowheads in the antisense-staining (**a2**). In a high magnification image of the medulla, a row of Tf mRNA-positive cells is indicated with blue asterisks (**b**).

**Figure 6 metabolites-12-00594-f006:**
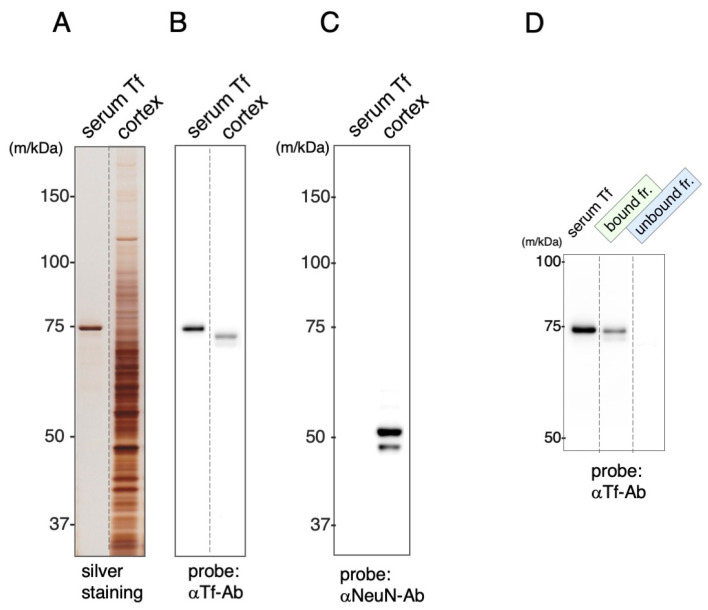
SDS-PAGE and immunoblotting of human brain extracts. Brain extracts were subjected to SDS-PAGE (**A**) followed by immunoblotting using anti-Tf antibody (αTf-Ab) (**B**). Another blot was probed with anti-NeuN antibody (αNeuN-Ab) (**C**). The extracts were separated into bound and unbound fractions by Tf antibody-immobilized beads and subjected to immunoblotting (**D**). Each lane separated on the gel is indicated by dotted lines.

**Figure 7 metabolites-12-00594-f007:**
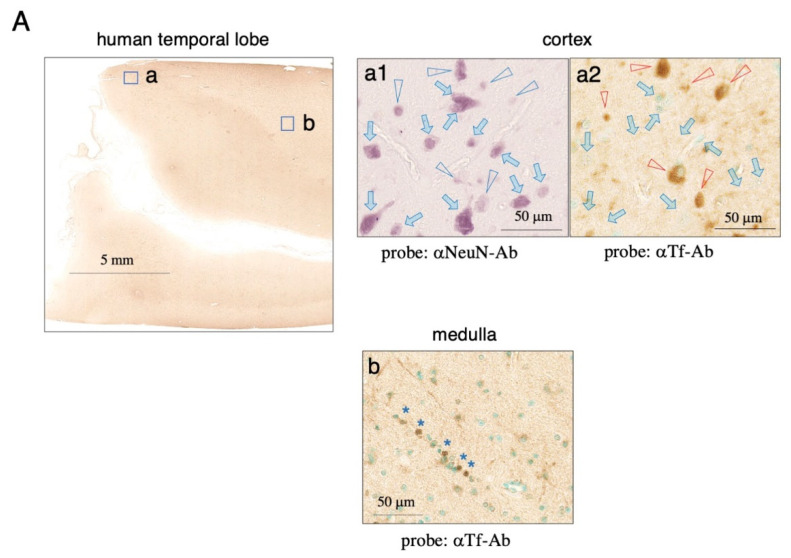
Immunohistochemistry using anti-Tf antibody for human temporal lobe. Sections of human temporal lobe were stained with anti-human Tf antibody and methyl green (**A**). Pairs of mirror-image sections from the cortex were prepared, with one section stained with anti-NeuN antibody for detecting neurons (**a1**), and the other with anti-human Tf antibody and methyl green for nuclear staining (**a2**). NeuN-positive cells are indicated with blue arrowheads (**a1**), while the same positions on the mirror-image section stained with anti-Tf antibody are indicated with red arrowheads (**a2**). NeuN-positive but Tf-negative neurons are indicated with blue arrows (**a1**,**a2**). In a high magnification image of the medulla, Tf-positive cells forming a row are indicated with blue asterisks (**b**).

**Figure 8 metabolites-12-00594-f008:**
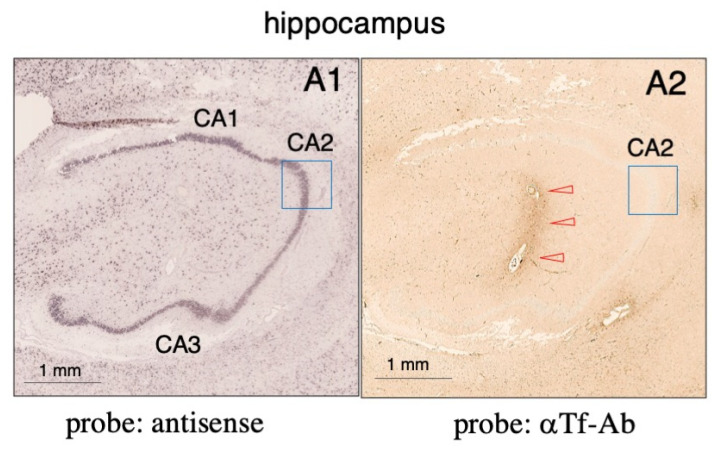
In situ hybridization and immunohistochemistry of human hippocampal tissue. Pairs of mirror-image sections of human hippocampus were prepared: one was subjected to in situ hybridization (**A1**), and the other was stained with anti-human Tf antibody and methyl green (**A2**). Immunostaining revealed diffuse signals surrounding blood vessels (**A2**, red arrowheads). In high magnifications of mirror-image sections, positive antisense signals are seen in the CA2 region (**B1**) along with anti-human Tf antibody immunostaining of the same area (**B2**).

## Data Availability

All data are contained within the article.

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
