# Peer review of "Expression of Transferrin Protein and Messenger RNA in Neural Cells from Mouse and Human Brain Tissue"

_metabolites, 2022, doi:10.3390/metabo12070594_

Round 1

Reviewer 1 Report

Researchers present a well-written manuscript characterizing the cellular level expression of transferrin in mouse and human brain. They find evidence of widespread transcription using ISH with limited protein immunoreactivity and discuss potential explanations. This work is complementary to the authors' own previous work characterizing post-translationally modified forms of transferring ostensibly originating from discrete organ systems including the brain, and its potential relevance for debilitating diseases such as Alzheimer's. Overall, this is a very nice paper and highly relevant for the special issue. A variety of minor issues and critiques should be considered:

Typographical errors: "SDA-page" is mistakenly listed instead of SDS page in a couple of locations.

What is meant by "mirror image sections" (e.g., line 199 and elsewhere)? Are these adjacent sections or are the same sections stripped and re-probed/antibody labeled on the reverse part of the section? The thickness of the sections should be listed in the methods. 

It would be helpful if the details of the performance of the commercial ELISA were listed including things such as limit of detection and coefficient of variation. 

In the main text and in the discussion, the explanation for the apparent discordance between the mRNA levels and protein levels in the tissue was given as high protein turnover or reduced translational efficiency. It seems to me that there could be more options. For instance, is it possible that the transcripts are transported (to axon terminals, or via exosomes) for local translation elsewhere? Are there splice variants that make the protein undetectable by the antibody used? These are just examples that might not be relevant to Tf. But there should be more discussion of this as it is central to your manuscript. At a minimum, please expand on the translational efficiency possibility. 

The sentence on line 259 is confusing. Was "well" supposed to be "were"? I recommend saying "were positively correlated" provided that it is accurate. 

Reviewer 2 Report

Title: Expression of Transferrin Protein and Messenger RNA in Neural Cells from Mouse and Human Brain Tissue

Submitted to: Metabolites

I found the paper poorly prepared. Firstly, this manuscript is inconsistent, it is imperfectly organized and it contains a lot of inappropriate information. Secondly, some experimental data are missing although they are mentioned. This manuscript doesn`t seem to have scientific value, I did not find a deep and new insight in the present manuscript. Therefore, I recommend that great amendments are warranted. I explain my concerns in more detail below.

1.       Introduction doesn`t reveal information regarding Transferrin`s (Tf) role in the brain. This section must to significantly extended in order to justify the necessity of this study.

2.       13 pages of this manuscript don`t provide any explanation of the purpose to study and compare the Tf mRNA and Tf protein expressions in mouse and human brains. Also, the conclusion part is missing, which makes it even more difficult to understand the need and purpose of the study presented in this report.

3.       Discussion section is not related to the research reported in the current manuscript. This section lacks evaluation, analysis, and discussion of data presented in the Result section. Instead, author discusses the difference in Tf expression within human and chicken brains. Despite the heterogeneity of the data and information presented, the Discussion section does not meet the criteria at all.

4.       Section 2.1. Expression of Tf mRNA in mouse brain doesn`t contain original Figure 1, it contains only a caption of Fig. 1.

5.       Section 2.2. Expression of Tf protein in mouse and human brain tissue. This section doesn`t contain any information or experimental data regarding Tf protein expression in the human brain. Some data should be added or the heading changed, in order to avoid misinformation.

6.       Quantification data of immunohistochemistry and blots must be presented along with the images.

7.       Section 2.6. Quantification of Tf protein in human brain tissue. It doesn`t contain any Tf protein data obtained by ELISA presented in graphs, charts, etc. The Figure and its caption, related to this section are missing. Tf protein detection by ELISA methodological part is not presented in the Materials and Methods section as well.

8.       Lines 117-118 contain unfinished sentence, that confuses.

9.       In Fig. 2 Tf mRNA hybridization with DIG-labelled antisense is impossible to distinguish from methyl green staining. DIG-labelled antisense color is not specified.

10.    In the caption of Fig. 4. LV and DG abbreviations are not clarified.

Reviewer 3 Report

Dear Authors, 

This is an interesting paper studying the expression of transferrin in brain. I have some methodological questions: 

1. How many mice did you use? This is important to define in the material and methods section, and also, are the pictures a representative pictures from the same mouse or different mouse? 

2. Why did you use only one human brain sample? Do you have access to more samples? With one sample it won't be possible to extract appropriate conclusions. 

3. Won't it be easier to analyze the expression of Tf RNA by Quantitative-PCR? 

4. I am missing a graph/table with the ELISA results of protein levels. 

5. The discussion is so short and it should address what the importance of this work is, correlate with other works,... The last paragraph referred to AD but you did not mention in the introduction. Please, include the relevant information in the introduction. 

Thank you very much. 

Reviewer 4 Report

Abstract: While the abstract is well written, I’m not sure it explains the motivations for examining regional expression patterns of transferrin (Tf) in the brain, namely that glycan isoforms thought to be derived from specific brain tissues are found in cerebrospinal fluid (CSF), and that these may have valuable applications as disease biomarkers. This was well explained in the introduction, but the abstract would benefit from a sentence explaining this rationale.

Introduction: The Introduction is well written and covers all the essential background material. Moreover, the authors demonstrate their rationale for performing this study as building on their previously published work on these glycan isoforms in CSF derived from choroid plexus and cerebral cortex.

Results and Figures: Unfortunately, it looks like Figure 1 is missing from the manuscript. Sadly, I cannot review these materials. If the Figure can be provided at a later time, I am happy to re-review, but I cannot recommend acceptance without seeing this figure.

It is an interesting result to find mRNA expression in the hippocampal neurons, but almost no protein. The translational or post-translational mechanisms underlying this result would be an interesting avenue of future study.

Section 2.2 is title “Expression of Tf protein in mouse and human brain tissue”. However, I cannot find any data in this section related to human brain tissue. Only mouse tissues are mentions in 2.2 as well as Figures 3 and 4.

I think it is important to see a visualization of the ELISA data presented in 2.6. Or perhaps a table. Something to draw attention to the data other than casually mentioning it in the text. Also, could the authors explain what they mean by normalized the signal to NeuN? The data as presented implies that the signal was actually normalized to total protein, and this is reflected in the methods as well under Section 4.1.

Material and Methods: One major issue noted from the methods is the lack of demonstrable reproducibility. The human brain tissue was from a single individual (a male). And for the mouse brains, the biological sample size is not specified. It is understandable why human tissue was limited, but replication with mouse tissues is necessary. Additionally, they used exclusively female mice for this study, and I do not see a good reason to do so. Although unlikely in this case, there is always the possibility that certain findings could be sex-specific, and it needs to be ruled out. Confirmation of their cell-specific findings with some modern approaches, such as single cell transcriptomics, would be nice, however it is understandable that these technologies are not yet accessible to all labs and institutions.

Discussion: The discussion is interesting and well written, especially regarding how their findings may relate to Alzheimer’s progression. However, the interesting aspects of the discussion seem to mainly surround their previous work, and it is difficult to see how much this particular manuscript adds much new information. It may not be sufficient for publication.

Round 2

Reviewer 2 Report

My comments have been addressed sufficiently, and it is ready for publication.

Thanks

Reviewer 3 Report

Dear Authors, 

Thank you very much for answering my questions and following my advice. 

Reviewer 4 Report

So from my original review, my main issue with the manuscript was reproducibility since the way the methods were written implied only female mice (an unspecified number) were used and a single human brain sample. The authors addressed this concern by revising the methods to more thoroughly describe the samples and sources. For the human samples, they actually had samples form multiple individuals pre-embeded in paraffin and the single individual (51 year old car accident victim) was the only freshly procurred sample. Comparing the original and revised manuscripts, it is clear how they were attempting to explain this and once they clarified with more detail, I feel more confident in the results.

For the mice, they said female C57 mice were used. However, in the revision they say 2 males and 3 females. I will trust that this was a mistake in writing and that they did actually analyze male and female samples and found no observable sex-dependent variability.